# Heat Shock Protein 70 (HSP70) Induction: Chaperonotherapy for Neuroprotection after Brain Injury

**DOI:** 10.3390/cells9092020

**Published:** 2020-09-02

**Authors:** Jong Youl Kim, Sumit Barua, Mei Ying Huang, Joohyun Park, Midori A. Yenari, Jong Eun Lee

**Affiliations:** 1Department of Anatomy, Yonsei University College of Medicine, Seoul 03722, Korea; jongyoulkim@yuhs.ac (J.Y.K.); sumit@yuhs.ac (S.B.); mayhuang@yuhs.ac (M.Y.H.); jhpark922@yuhs.ac (J.P.); 2BK21 Plus Project for Medical Science and Brain Research Institute, Yonsei University College of Medicine, 50-1 Yonsei-ro, Seodaemun-gu, Seoul 03722, Korea; 3Department of Neurology, University of California, San Francisco & the San Francisco Veterans Affairs Medical Center, Neurology (127) VAMC 4150 Clement St., San Francisco, CA 94121, USA

**Keywords:** heat shock protein 70, brain injury, chaperone neuroprotection, pharmacological induction

## Abstract

The 70 kDa heat shock protein (HSP70) is a stress-inducible protein that has been shown to protect the brain from various nervous system injuries. It allows cells to withstand potentially lethal insults through its chaperone functions. Its chaperone properties can assist in protein folding and prevent protein aggregation following several of these insults. Although its neuroprotective properties have been largely attributed to its chaperone functions, HSP70 may interact directly with proteins involved in cell death and inflammatory pathways following injury. Through the use of mutant animal models, gene transfer, or heat stress, a number of studies have now reported positive outcomes of HSP70 induction. However, these approaches are not practical for clinical translation. Thus, pharmaceutical compounds that can induce HSP70, mostly by inhibiting HSP90, have been investigated as potential therapies to mitigate neurological disease and lead to neuroprotection. This review summarizes the neuroprotective mechanisms of HSP70 and discusses potential ways in which this endogenous therapeutic molecule could be practically induced by pharmacological means to ultimately improve neurological outcomes in acute neurological disease.

## 1. Introduction

After various insults to the brain, a coordinated stress response which seems to protect it from further injury occurs. Heat shock proteins (HSPs) are the most exhaustively studied stress proteins. They were originally noted when sublethal heat stress was applied to cells. Postmortem studies have also documented induction of HSPs in the human brain following different types of thermal stress, such as hyperthermia or fire-related fatalities [1]. When core body temperature exceeded 40 °C, increased transcripts of HSPs were detected in postmortem brain specimens. The study of autopsied brain specimens of patients who had suffered from hyperthermia also led to the conclusion that HSP70 induction could be a brain biomarker of death [2]. HSPs are chaperones that typically act within the cytosolic space, engaged in assisting with protein folding, degradation, complex assembly, and translocation. They have demonstrated the ability to inhibit the accumulation of damaged proteins as well as to facilitate the construction of polypeptides of newly synthesized proteins. The diverse roles by which HSP70 and HSP90 regulate aggregated proteins appear to be involved in neuroprotection as demonstrated by models of brain injury. HSP70 induction also represents an endogenous protective mechanism that occurs in the penumbra of the hippocampus, but not of other core areas, in the ischemic stroke model [3,4]. More than two decades of research involving such models have shown that HSP70 has the ability to protect against multiple types of cell death, including apoptosis and necrosis. Specifically, HSP70 interferes with multiple cell death pathways [5,6]. HSP70 also modulates inflammatory pathways and, thus, appears to improve neurological outcomes through interrupting both cell death and immune responses [7]. It should be noted, however, that these studies possessed limited translational utility because they relied upon either genetic mutant models or gene transfer models, and upon heat stress to induce HSP70 overexpression. In the HSP70 research trajectory, multiple disciplines have studied geldanamycin (GA) and 17-allyamino-demethoxygeldamycin (17-AAG), which block HSP90 leading to the induction of HSP70 [8,9]. The possible clinical applications of HSP70-inducing pharmacological compounds in neuroprotective therapies for ischemic stroke and associated conditions warrant further research [8]. Here, we discuss the mechanisms of HSP70 neuroprotection in brain injury (ischemic stroke and traumatic brain injury (TBI)), along with pharmacological HSP70 inducers and their possible applications at the clinical level.

## 2. Classification and Functional Role of Heat Shock Protein 70

At the onset of brain injury (for instance, of ischemic stroke or TBI), the synthesis of most cellular proteins is downregulated. However, HSPs belong to a small class of proteins that are, instead, upregulated, and have been collectively referred to as stress proteins. HSPs are classified in accordance to their molecular mass. Constitutive HSPs, such as HSP90, HSP40, and HSP70, have housekeeping functions within the cell [10]. HSP70 and HSP90 are two highly conserved ATP-dependent HSPs that modulate unfolded proteins.

HSP90 is an ATP-dependent chaperone associated with protein homeostasis [11]. It is required for the homeostasis of a number of key cellular proteins and protein complexes. HSP90 client proteins belong to distinct functional classes, such as transcription factors (e.g., HIF1α, ATF3, and p53), steroid hormone receptors (e.g., estrogen receptor, glucocorticoid receptor, and progesterone receptor), and kinases (e.g., EGFR, B-raf, and SRC). HSP90 and cochaperones bind to client proteins in an ordered pathway that involves sequential ATP-dependent interactions of the client protein with HSP70 and HSP90 [12]. Cochaperones are fundamental in regulating the ATP enzymatic activity of HSP90 in the cytoplasm and in mediating interactions between HSP90 and substrate [12,13]. They regulate the function of HSP90 by either inhibiting or activating the ATPase of HSP90 and by recruiting specific client proteins in different ways [14,15]. HSP90 directs the folding and activation of a wide variety of substrate proteins, most of which are kinases and transcription factors involved in signal transduction and regulatory processes [16,17]. Furthermore, many diverse pathological conditions such as cancer, neurodegenerative diseases, and infectious diseases involve HSPs. Of note, HSP90 functions in tandem with many additional chaperones, including HSP70 and HSP40, and also with cochaperones, including those containing tetratricopeptide repeat (TPR), to refold many denatured proteins [18].

HSP70 exists in several forms, which are either constitutive or inducible. Heat stress is an HSP70 catalyst for brain cells; it produces a marked expression of induced HSP70 family chaperones [19]. The neuroprotective properties associated with HSP70 have been the most comprehensively studied. HSP70 family chaperones exist in a multitude of different forms, including cytosolic, HSP73 (also Heat shock cognate [HSC]70); inducible cytosolic, HSP72; mitochondrial, HSP75/mortalin; and ER, HSP78/BIP. Other variants of HSP70 include HSP72 and HSP70i. HSP70 reacts to hydrophobic peptide segments, and these reactions are ATP-dependent. In HSP70, a C-terminal substrate-binding domain works to identify unstructured polypeptide segments, while a N-terminal ATPase domain is involved in HSP70’s protein folding functions. For the protein folding role, HSP70 fluctuates between ATP-bound open and closed states with low and high substrate affinity, respectively [20].

The chaperone functions of HSP70 and HSP90 work in a coordinated manner to improve denatured protein stability [21]. In contrast to the refolding function of these proteins, when HSP90 is blocked by pharmacological substances such as GA or 17-AAG, it rapidly degrades client proteins through the ubiquitin–proteasome pathway [22]. Moreover, HSP70 and its cochaperone HSP40 assist in the degradation of various proteins [21].

## 3. Mechanism of Heat Shock Protein 70 Induction and Its Interactions with Cochaperones and Client Proteins

HSP transcription is regulated by transcription factor-heat shock factors (HSFs), which are translocated into the nucleus where they interact with conserved heat shock elements (HSEs) to upregulate genes that code for the induced HSPs. HSEs are generally located in the upstream untranslated region of HSFs target genes. HSF activation is accomplished at the level of protein–protein interaction and posttranslational modification [23]. Conversely, some HSFs expressed through HSPA1A/B/L (HSP70), HSPA1A (HSP72), and HSPC1 (HSP90), directly interrupt HSFs through binding to its trimerization domain. In addition, a cytoplasmic histone deacetylase and a valosin-containing protein lead to the suppression of the HSF–HSP complex [24].

Ischemia and other conditions, which are associated with unfolded proteins, lead to the dissociation of HSFs from HSPs. Dissociated HSFs migrate to the nucleus of the stressed cell, where they become phosphorylated, often by a protein such as kinase C. This process forms activated trimers that bind to the HSE, which is a highly conserved regulatory sequence located on the heat shock gene. After binding to HSE, HSFs then bind to the HSP70 gene’s promoter region, which leads to more HSP70 generation [8]. HSP70 production is influenced by HSP90, because the latter binds to HSF1. HSF1 can be freed to bind to HSEs in instances where HSP90 dissociates from it, thus facilitating further HSP70 induction [25].

The ATPs, HSP40 and HSP90 induce HSP70 synthesis and act as the molecular chaperones of damaged cells by repairing their denatured proteins. This role is completed through refolding and trafficking of the denatured proteins in the damaged cells, ultimately resulting in a multiple iteration of protein refolding. HSP40 binds to HSP70 with high affinity and is responsible for the ATPase activity of HSP70 and for maintaining it in an ADP-bound state. HSP40, by itself, cannot stabilize HSP70’s ADP state. Due to this, and because of newly minted HSP70-substrate complexes, early dissociation occurs [26]. Thus, there are additional cochaperones involved. HSP70 interacting protein (HIP) binds to the HSP70 ATPase domain, thus maintaining HSP70’s ADP-bound state, and protects the chaperone–substrate complex from dissociating [26]. Another cofactor, HSP70/HSP90-organization protein (HOP), binds to the HSP70 C-terminus, allowing HOP to recruit HSP90 to the HSP70 substrate complex [27,28,29]. As HIP stabilizes the ADP/ATP exchange of HSP70, HOP’s interaction with HSP70 and HSP90 releases HSP70 chaperone components (Figure 1A). HSP70 utilizes its cochaperones, CHIP (C-terminus of HSP70/HSC70 interacting protein) and Bcl-2–associated athanogene (Bag)-1, indirectly for its role in protein degeneration (Figure 1B). CHIP, an ubiquitin ligase, interacts with HSP70 by rounding unfolded proteins and then ubiquitinating them, thus facilitating the degradation of seized proteins [30,31]. Under stress conditions, HSP70 induction is improved by CHIP function. Moreover, CHIP regulates HSP70’s development in the stress recovery process; it probably achieves protein homeostasis by regulating chaperone levels while in stress and recovery processes [32]. Bag-1, an HSP70 cochaperone, is also associated with the proteasomal degradation of abnormal proteins. Bag-1 and CHIP’s interaction and co-expression directly enhance the model chaperone substrate degradation [33]. They work together to update the chaperone systems from a protein folding role to a degradation function. HSP90 is interlinked with HSP70 regarding protein degradation. Both chaperones transitorily associate with shared cochaperones. HSP90 transfers client proteins to HSP70 via a transient interaction. CHIP, that is associated to HSP70, ubiquitinates and pushes client proteins to proteasome mechanisms (Figure 1B) [34,35,36]. Thus, HSP90 is an inhibitor to substrate ubiquitination and degradation. In contrast, HSP70 is a promoter of ubiquitination and degradation [37,38].

## 4. Role of Heat Shock Protein 70 in Brain Injury

Injury to the brain either from loss of blood flow or direct trauma to the brain evokes a complex multicellular pathophysiological state. Such injuries lead to different types of cellular damage, ranging from acute excitotoxic stress to instances of delayed programmed cell death. Inducible HSP70 levels are low while in homeostasis; however, injury drastically raises its expression. Therefore, HSP70 is demonstrated to be a marker of stress in cells. HSP70 induction was originally studied in a stroke model; within this model, HSP70 was initially observed in neurons of the brain regions that appeared relatively resistant to injury [39,40]. It was also detected in brain regions surrounding the infarct (penumbra) [41]. Later, HSP70 was observed in endothelial cells and glia, such as astrocytes and microglia [41]. In particular, HSP70 was detected in the brain regions surrounding the area of stroke (penumbra), but was absent in the areas most affected by the infarct (core) [42]. However, HSP70 mRNA was still detected within this core [41]. Thus, it was concluded that HSP70 was mainly detected in cells that survived the ischemic insult, while brain areas that failed to translate HSP70 remained vulnerable to ischemic injury. Similarly, in a model of global cerebral ischemia, which attempts to recreate ischemic brain injury due to cardiac arrest, HSP70 was mainly observed in the cornu amonis (CA) 3 and the dentate granules of the hippocampus that tend to survive the ischemic episode, whereas HSP70 was not induced in CA1 neurons [43]. Equally to that observed in stroke models, HSP70 mRNA was detected in all ischemic hippocampal regions, including CA1 neurons. However, CA1 neurons, which tend to be more vulnerable to ischemic insults compared to neurons of other hippocampal regions, failed to express the protein [44]. These observations generated debate as to whether HSP70 expression was an epiphenomenon of the injury or facilitated cell survival.

Experimentation demonstrating that HSP70 overexpression using viral vectors improved survival of neurons and astrocytes in stroke models finally provided direct evidence of a neuroprotective role of HSP70 [45,46]. Similarly, transgenic mice that overexpress HSP70 have smaller lesion sizes and better neurological outcomes, whereas a deficiency exacerbates lesions and poor outcomes [47,48]. Although less studied compared to brain ischemia, HSP70 was found to exhibit similar patterns in experimental models of TBI [7]. Many mechanisms leading to this protective effect have since been demonstrated in stroke and related models, ranging from prevention of protein aggregates to modulation of cell death pathways [20].

## 5. The Neuroprotective Effect of Heat Shock Protein 70 via Cell Death Pathways

HSP70 induction can lower protein aggregates and intracellular inclusions [49], but specific chaperone interactions seem to underlie other ways in which HSP70 may lead to neuroprotection. After brain injury, apoptosis occurs either via the intrinsic or mitochondrial pathway [50] or via the extrinsic or surface receptor-mediated pathway [51]. Recent research demonstrated that HSP70 impedes apoptosis via direct and indirect means (Figure 2).

The intrinsic apoptotic pathway initiates with the release of various factors from the cell’s mitochondria. In response to brain injury (and to the resultant oxidative stress), the mitochondria develop a permeability transition pore which leads to the discharge of cytochrome c to the cytosol, where a number of pro-apoptotic molecules ultimately causes the activation of effector caspases. Among these molecules are the Bcl-2 family members, some of which are pro-apoptotic. These molecules are primary regulators of the mitochondrial membrane. There are three subgroups classified by their structural homology. The first group includes the anti-apoptotic proteins Bcl-2, Bcl-XL, and Bcl-w. The second group includes the pro-apoptotic proteins Bax and Bak. The third group includes the BH3-only proteins Bad, Bid, Bim, Noxa, and PUMA. Bcl-2 family proteins conduct multiple functions during brain injury. BH3-only proteins, via interactions with Bcl-2 family members, assist with neuronal cell death after ischemic stroke [52]. Mitochondria-based apoptosis is related to the apoptosome. This occurs when procaspase-9 interacts with apoptosis protease activating factor-1 (Apaf-1) in the cytosol and leads to activation when cytochrome c is translocated to the cytosol from the mitochondria. Bcl-2, an antiapoptotic protein, interrupts cytochrome c release. Furthermore, caspase-9 activation induces the activation of a multitude of effector caspases such as caspase-3. In neuronal stem cells, induction of HSP70 by recombination plasmid pEGFP-C2-HSP70 significantly blocks caspase-3 and reduces neural cytotoxicity, including neuron loss and synapsis damnification, in cocultured cells [53]. After an inflammatory stimulus, oligodendrocyte precursor cells and mature oligodendrocytes of HSP70 deficient mice enter apoptosis caused by caspase-3 activation [54]. Caspase-independent pathways have been described to cause apoptosis via apoptosis inducing factor (AIF) [55]. AIF translocates into the nucleus from the mitochondria. Once in the nucleus, AIF can trigger apoptosis in the absence of caspases. Like cytochrome c, AIF release can also be prevented by Bcl-2 [56]. Second mitochondria-derived activator of caspases (Smac)/direct inhibitor-of-apoptosis protein (IAP) binding protein with low pi (Smac/DIABLO) is also a regulator of apoptosis. These are also discharged from the mitochondria and impede apoptosomal inhibition by IAPs. HSP70 seems to affect multiple aspects of the intrinsic apoptotic pathway. HSP70 interacts with components of the apoptotic machinery both upstream [57,58] and downstream of mitochondrial events [59]. In experimental stroke models, HSP70 interfered with cytochrome c release [43,60] and inhibited AIF translocation to the nucleus [61] while reducing ischemic brain injury. It lowers the recruitment of procaspase-9 into the apoptosome, as observed in HSP70 overexpressing transgenic mice, and sequesters AIF [62]. HSP70 also prevents the release of proapoptotic protein Smac/DIABLO from myocyte mitochondria [63]. Mitochondrial HSP70/HSP75/mortalin assistants act to maintain mitochondrial membrane potential, which could be beneficial to mitochondrial function and mitochondrial protein import [64]. Astrocytes with HSP induction showed reduced cell death following in vitro stroke models with preserved ATP [65]. The above outcomes are connected to lowered reactive oxygen species (ROS) creation, and maintained mitochondrial membrane potentials [66] and glutathione levels [67]. Overexpression of HSP70 has been demonstrated to elevate mitochondrial antioxidant enzyme activity in myocardial cells [68]. Bcl-2 is a critical component in hindering the onset of apoptosis due its ability to block the release of cytochrome c and AIF, and viral vector-mediated HSP70 overexpression was linked to elevated Bcl-2 protein levels in hippocampal neurons [69]. HSP70 has also been shown to reduce heat-induced apoptosis by preventing the migration of the pro-apoptotic Bcl-2 family member Bax, and thus blocking the mitochondrial discharge of pro-apoptotic factors [58]. Previous studies of HSP27 and its anti-apoptotic activity have established that stress-induced Bax oligomerization and migration to the mitochondria can be suppressed indirectly by HSP27 [70]. HSP27 has also been shown to phosphorylate the survival kinase Akt/protein kinase B (PKB) [71] and to inactivate prodeath c-Jun N-terminal kinase (JNK) [72]. HSP70 also interferes with the activity of Apaf-1, a critical component for apoptosome formation and activation of caspase-9 [62], although others have failed to show such an interaction [57].

The extrinsic or cell surface-mediated pathway of apoptosis involves interactions with death receptors found on the plasma membrane. This pathway is also known as the “death receptor pathway”. Death receptor ligation activates caspase-8 and caspase-10. This has the potential to activate effector caspase-3 [73]. The TNF family of ligands, including FasL, TNF, CD40L, and TRAIL, promotes the activation of many death receptors. Many of these ligands are released extracellularly as part of the ischemic immune response [74]. One of the first death receptors identified was Fas, which activates when it binds to its ligand FasL. When FasL binds to Fas, the cytoplasmic adaptor protein Fas-associated death domain protein (FADD) is recruited to this complex. FADD possesses a “death effector domain” at the N-terminus that can bind to procaspase-8 [75]. This is known as the death-inducing signaling complex (DISC). The proteolytic cleavage, which causes transactivation of procaspase-8 to caspase-8, is catalyzed by the DIAC [75]. Activated caspase-8 ultimately leads to the activation of caspases-3 and -10 [76].

HSP70 has been demonstrated to engage with extrinsic death receptor signaling pathways. HSP70 is known to bind to cell surface receptors TRAILR1 and TRAILR2, and to death receptors 4 (DR4) and 5 (DR5), where they bind to a cytokine called TRAIL to induce TNF-related apoptosis. Therefore, the TRAIL-induced assembly and activity of the DISC becomes inhibited [77,78]. HSP70 can neutralize Bid activation and the following apoptosis once DISC formation has occurred and caspase-8 has been activated [79]. HSP70 has been demonstrated to interact with the Fas pathway. Dynamin, a molecule typically associated with synaptic transmission, is also known to traffic Fas to the cell’s surface from the Golgi apparatus [80]. Fas binds to FasL when it translocates to the cell surface, activating caspase-8 and, therefore, leading to cell death. We recently showed that HSP70 inhibits Fas trafficking to the cell surface via its interaction with dynamin [48]. Thus, HSP70 also prevents the extrinsic or receptor-mediated apoptosis with specific chaperone interactions.

Several studies have shown that microRNAs (miRNAs), which interact with multiple target messenger RNAs (mRNAs), coordinate the regulation of target genes. Several miRNAs such as miRNA-1, miRNA-21, and miRNA-24 may contribute to an increased expression of several cytoprotective proteins, although no targets have been validated. Some studies showed that muscle-specific miRNA-1 levels can change in the condition of ischemic myocardium [81,82,83], and that two of miRNA-1’s targets are HSP60 and HSP70 [84]. Recently, Ouyang et al. showed that miRNA-181a, which is expressed at high levels in the brain, regulates HSP70 family chaperones and the outcome of ischemic stroke [85]. Mutual expression of miRNA-181a and HSP78/BIP was found both in the core and the penumbra in an ischemic stroke model. miRNA-181a mimic reduces and its inhibitor/antagomir enhances HSP78/BIP expression [85]. miRNA-181a also targets Bcl2, which is an antiapoptotic protein [86] and has been shown to be upregulated by HSP70 overexpression [69]. miRNA-181 can also target HSP78/BIP and potentially target the 3′ UTRs of HSP72 and HSP75/mortalin. Following ischemic stroke, miRNAs could conceivably target multiple chaperones and efficiently modulate cell death.

## 6. Inflammation Regulation of Heat Shock Protein 70

Inflammation of the central nervous system (CNS) is a feature of many acute neurological injuries, including brain trauma, stroke, and other cerebral hypoxic-ischemic injuries [87]. Brain injury elicits an inflammatory reaction beginning with the activation of endogenous microglia and with peripheral leukocyte influx into the cerebral parenchyma [88,89]. Upon inflammatory cell activation, cytotoxic agents such as some cytokines, which are increasingly viewed as key contributors to ischemic cell death [90], are discharged. Some studies indicate that nonimmunologic brain cells, such as astrocytes and even neurons, can elaborate these same inflammatory molecules. These inflammatory responses are thought to exacerbate brain damage, thus presenting a major opportunity for potential treatments. HSP70 has been demonstrated to possess a modulating role in regulating immune responses in cases of brain injury. HSP70 has been demonstrated to control inflammation both inside and outside the cell. In the intracellular setting, HSP70 appears to inhibit pro-immune responses; whereas in the extracellular setting, it seems to do the opposite and potentiate such responses.

HSP70 is capable of interacting with transcription factors, including those which trigger inflammations, such as the nuclear factor-kappaB (NF-kB, which consists of the heterodimers p65 and p50). One study in astrocytes showed that HSP70 was capable of reducing NF-kB activation [91]. HSP70 can also interact with immune molecules themselves, such as matrix metalloproteinases (MMPs) and ROS; it seems to limit inflammatory responses in such cases. Increased HSP70 within the cell has been shown to decrease the production of nitric oxide and inducible nitric oxide synthase (iNOS) in inflammatory cells. Heat stress is also associated with the reduced secretion of tumor necrosis factor-alpha (TNF-α) as well as with diminished generation of ROS. HSP70 has been shown to inhibit the cellular responses to inflammatory cytokines including TNF-α and interleukin (IL)-1 [92,93], and increased HSP70 in macrophages prevents lipopolysaccharide (LPS)-induced increases in TNF, IL-1, IL-10, and IL-12 [93,94]. In a model of intracerebral hemorrhage, HSP70 also reduced TNF-α while attenuating blood brain barrier (BBB) disruption and brain edema, and enhanced neurological function as well [95].

Induction of HSP70 in phagocytes through heat shock decreased nicotinamide adenine dinucleotide phosphate hydrogen (NADPH) oxidase activity in neutrophils and increased superoxide dismutase, a superoxide scavenger [96]. Recently, we demonstrated that HSP70 induction by heat stress halts IkB, JNK, and p38 phosphorylation. HSP70 also seems to inhibit the binding to DNA of several transcription factors, including NF-kB, activator protein-1 (AP-1), and signal transducer and activator of transcription factor 1 (STAT-1), with subsequent inhibition of their pro-inflammatory transgenes [97]. Other studies have also demonstrated that prior-heat stress, due, in part, to HSP70 overexpression and prevention of nuclear NF-kB translocation, interferes with these inflammatory responses [98,99]. When ischemic stroke induces generation of cytokines, such as TNF-α that can activate NF-kB–dependent gene transcription through the NF-kB pathway, the inhibitor of kB (IkB) acts as another target of HSP70 and, when bound, can interrupt NF-kB activation by preventing phosphorylation of IkB (Figure 3) [91,92]. Some studies have also shown that HSP70 interacts with NF-kB and/or its regulatory proteins [100,101], but the mechanisms may depend on the nature of the stimulus. In a cell death model induced by TNF-α, HSP70 directly inhibited IkB kinase (IKK) activity [102], while in a stroke model, HSP70 seemed to interact with both NF-kB and IkB, ultimately leading to reduced IkB phosphorylation by IKK [100]. NF-kB inhibition by HSP70 further prevented transcription of immune genes and led to a neuroprotective effect.

NF-kB regulates many pro-inflammatory genes, including matrix metalloproteinase (MMP)-9. Our group showed that MMP-9 mRNA was reduced when HSP70 was overexpressed in astrocytes later exposed to ischemia-like injury [46]. Furthermore, HSP70 in astrocytes suppresses MMP-9 protein expression [46]. MMP-2, while not regulated by NF-kB, was also decreased in an HSP70 overexpression context, which suggests that HSP70 inhibits other transcription factors as well. In fact, studies in heat stressed alveolar macrophages [18] and primary astrocytes [97], showed that HSP70 interfered with STAT-1, which is known to regulate MMP-2 expression [103]. Additionally, HSP70 seems to inhibit the generation of activated MMPs, as evidenced by a reduction in cleaved forms of several MMPs. In a model of TBI, HSP70 overexpression inhibited, while HSP70 deficiency exacerbated, brain hemorrhage, and this was correlated with changes in MMP levels and activity [7]. Thus, there are several ways in which HSP70 inhibits the post injury immune response.

Extracellular HSP70 also has the capacity to modulate immune responses [104]. HSP70 has been shown to accomplish opposite roles depending on the type of insult and the immune response that follows. Extracellular HSP70 appears to potentiate adaptive immune responses, and has been widely studied. Complexes of HSP70 and peptides trigger CD8+T-cell responses [105]. Injecting mice with these HSP70–peptide complexes leads to similar responses, and suggests that HSP70 acts as an adjuvant [106]. In the extracellular environment, HSP70 can also potentiate antigen presentation, and reacts with macrophages and dendritic cells via CD40, CD91, and LOX-1 receptors [107]. Extracellular HSP70 has also been shown to elicit innate immune responses by interacting with Toll-like receptors (TLRs), causing subsequent NF-kB activation and target gene upregulation, including iNOS and pro-inflammatory cytokines [108]. HSP60 and HSP90 appear to interact with TLR-2 and -4 [109]; however, these results have been controversial, as some preparations of recombinant HSPs may have contained endotoxin, the classic ligand for TLR-4 [110].

## 7. Heat Shock Protein 70 as a (Pharmacological) Therapeutic Target for Brain Injury

Understanding the mechanisms by which HSPs function within the cell under conditions of stress and injury may reveal potential therapeutic targets. A few studies have attempted a direct delivery of HSP70 to the brain, while others have employed the cell’s endogenous capacity to upregulate HSP70 through its coordinated actions with HSP90. After brain injury, treatments to increase HSP70 in the cell might be a worthwhile approach to treat neurological conditions associated with inflammation and cell death. A few pharmacological approaches to delivering HSP70 have been explored. In one study, exogenous HSP70 was conjugated to the HIV TAT protein in order to improve brain delivery following intravenous administration. HSP70-TAT reduced infarct volumes, improved neurological outcomes, and led to higher survival of neural progenitors in a stroke model [111]. Similar observations were found when recombinant HSP70 was administered in a rodent stroke model [112]. In this latter study, HSP70 was conjugated with Fv, which is a single chain fragment of the anti-DNA antibody mAb 3E10 that allows for penetration into the cell, and the Fv fragment was used as a delivery vehicle. Interestingly, systemic Fv-HSP70 treatment seemed to enter the ischemic side of the brain but not the contralateral nonischemic side, which suggests that it may bind to a still unknown target within the injured brain.

Since HSP90 sequesters HSF1 in the cytosol and prevents transcription of target HSPs, including HSP70, the administration of pharmacological inhibitors of HSP90 is another strategy to endogenously increase HSP70 expression. A few studies have now explored the efficacy of pharmacological induction of HSP70 via HSP90 inhibitors in brain injury models. This was recently reviewed by us, and we refer the reader to the review for more details on this topic [8]. Such inducers have shown salutary effects against both local and global experimental cerebral ischemia [113,114]. Due to the HSP70 abilities to reduce apoptosis and inflammation after ischemic stroke, there is an obvious interest in the HSP70 induction capacity of HSP90 inhibitor agents. One such HSP90-antagonist, GA, has been studied in stroke models with salutary results [113], but has failed at the clinical level due to poor solubility in water [115] and to unacceptable renal and hepatic toxicity [116,117]. The GA analogue 17-AAG is less toxic and has since advanced to phase 3 clinical trials for cancer therapy [116,118]. Consistent with the findings seen with GA, 17-AAG treatment improved outcomes in experimental TBI, with a reduction in brain hemorrhage [119]. 17-(2-dimethylaminoethyl) amino-17-demethoxygeldanamycin (17-DMAG), with better solubility, was developed, and was found to decrease microglial activation and to inhibit phosphorylation of IκB resulting in reduced nuclear translocation of NF-kB (p65) in a stroke model [120].

There are other HSP70-inducing groups, such as the purine-based compounds and the resorcinols. Purine-based compounds were designed to resemble ansamycins; hence, they use ADP to bind to HSP90’s ATP binding site [121]. BIIB021 (also named CNF-2024) is also an HSP70-inducer and an HSP90-inhibitor. It is being studied in a phase 2 clinical trial using an oral form [118,122]. Another study in lymphoma cells demonstrated BIIB021 to be a potent anti-inflammatory compound that can suppress NF-kB [123]. What is still unclear is if these purine-based compounds accomplish the same roles in other types of cells, and whether they can penetrate the BBB. The resorcinol group includes radicicol based compounds, which bind to HSP90’s ATP-binding pocket [124,125]. Despite their HSP90-inhibiting properties, drug development of the resorcinol groups is slow because they have been found to easily degrade in vivo [118]. Some variants such as NVP-AUY922 and AT-13387 were designed to avoid this limitation, and have also been studied as potential anti-inflammatory compounds for cancer treatment. Resorcinol has yet to be studied in brain injury or inflammation [118]. There is, however, some work published about another HSP70 inducer, geranylgeranylacetone (GGA), known for its antiulcer properties. In a stroke model, GGA led to not only a neuroprotective effect but also to a reduced post stroke inflammation via HSP70 upregulation and protein kinase C (PKC) activation [126,127]. Others found that the administration of GGA also led to a neuroprotective effect in a TBI model by inhibiting microglial activation and reducing apoptosis of neurons [128]. Thus, promising preclinical studies of HSP90 inhibitors together with clinical experience in cancer patients may stimulate appropriate trials in stroke and TBI patients.

Other pharmacological inducers of HSPs have been studied in models of neurodegeneration and cancer, but they have yet to be tested in brain injury models. Myricetin, a flavonoid, was shown to enhance intracellular levels of HSF-1 and HSP70, and to suppress abnormal protein aggregation and remove various toxic neurodegenerative disease-associated proteins [129]. Celastrol, a quinone methide family member isolated from the Thunder God Vine, has been shown to upregulate HSF-1, which, in turn, leads to HSP induction; this has been shown in human neuronal cells [130]. Celastrol has been shown to protect motor neurons from kainic acid induced excitotoxicity while upregulating HSPs [131]. Celastrol has also been shown to improve outcome in a stroke model, but this study did not explore whether this was related to HSP induction [132]. In a metastatic cancer model, a recent study showed that electrochemotherapy with betulinic acid or with cisplatin increased HSP27 and HSP70, and they proposed that electric field stress combined with drug administration led to the induction of HSPs [133]. Thus, promising observations in related nervous system disease models suggest that these approaches should be tested in acute brain injury models.

## 8. Conclusions

The role of the HSP family as potential neuroprotectants for the treatment of acute brain insults has been increasingly recognized, along with a more detailed elucidation of the mechanisms underlying this beneficial effect. Several studies have now reported a neuroprotective effect, particularly after HSP70 induction or administration through pharmacological manipulations. HSP70 also appears to be accountable for multiple protective mechanisms as a molecular chaperone. The multifaceted mechanisms of protection by HSPs suggest that these may be an effective therapeutic target. While some HSP90 inhibiting compounds have already been investigated in cancer clinical trials, there are no studies in patients with neurological conditions. With the development of several pharmacological HSP70 inducers and of methods to deliver the protein itself, it seems that we may be ready for clinical trials.

## Figures and Tables

**Figure 1 cells-09-02020-f001:**
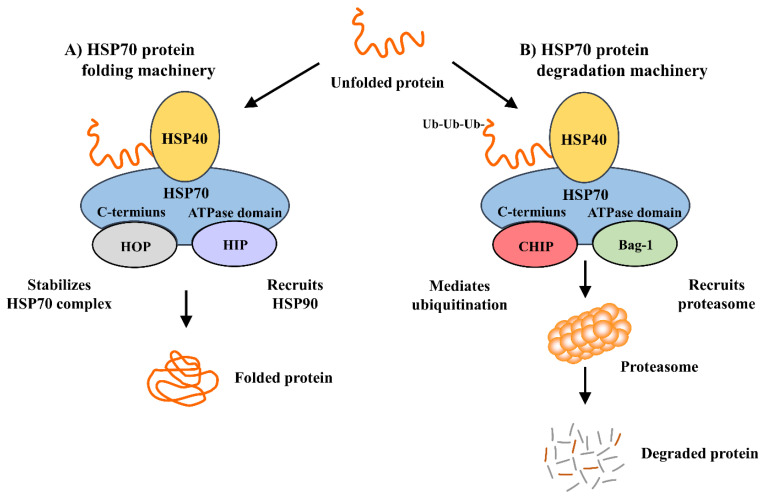
HSP70 chaperone machinery. HOP and CHIP compete for HSP70’s C-terminus during protein folding and degradation, while HIP and Bag-1 compete for the ATPase domain. Depending on the site of HIP or Bag-1 binding, HSP70 may lead to protein folding (HIP/HOP pathway; (**A**) or to protein degradation through the proteasome (CHIP/Bag-1 pathway; (**B**). Ub = ubiquitin.

**Figure 2 cells-09-02020-f002:**
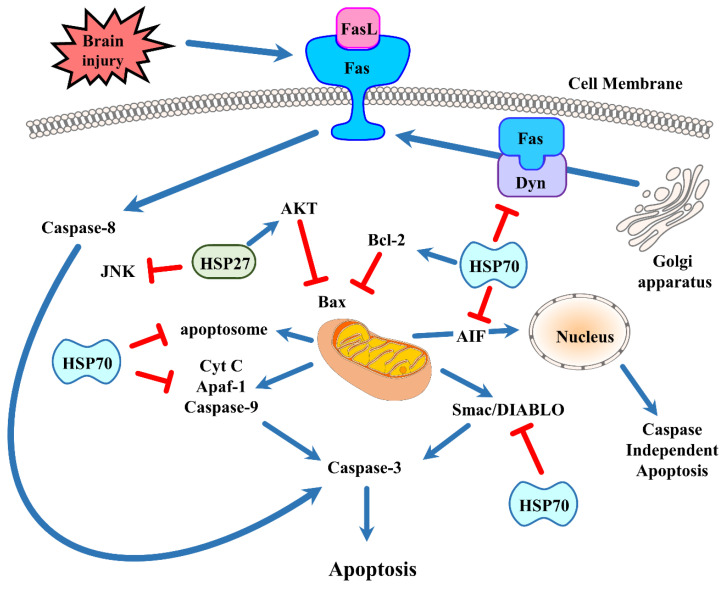
Brain injury induces apoptotic cell death by distinct pathways. The intrinsic pathway is centered around signals emanating from the mitochondria, whereas the extrinsic or receptor-mediated pathway begins when death receptors bind to their ligands. The prototypical death receptor Fas is shown here. HSP70 and HSP27 have been shown to interfere with apoptosis as indicated in the figure. See text for more details. (FasL = Fas ligand; AIF = apoptosis inducing factor; Akt = protein kinase B; Apaf-1 = apoptosis protease activating factor-1; Cyt C = cytochrome c; JNK = c-Jun N-terminal kinase; Dyn = dynamin).

**Figure 3 cells-09-02020-f003:**
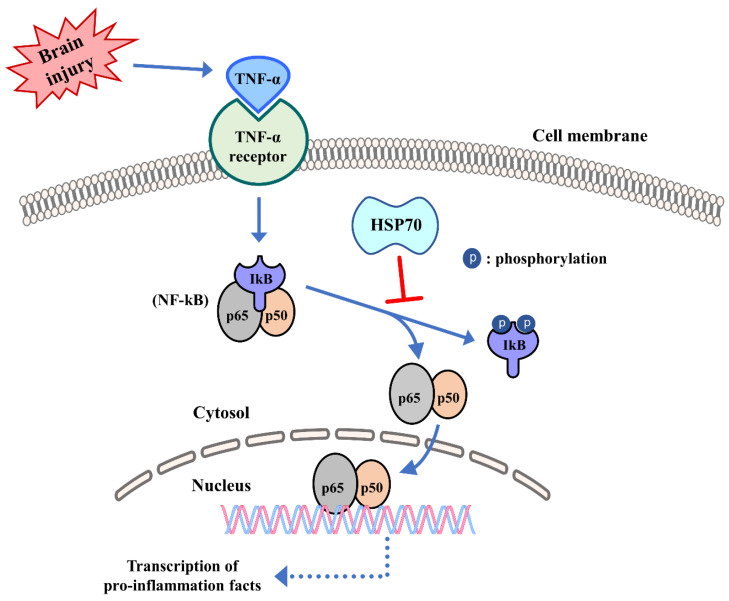
Influence of HSPs in innate immunity. Following ischemic stroke, HSPs have been shown to inhibit the activation of the transcription factor NF-kB and to prevent its nuclear translocation. Acute brain insults (brain injury) trigger activation of NF-kB by causing the phosphorylation and degradation of its inhibitor protein IkB, which normally keeps NF-kB (which consists of the heterodimers p65 and p50) tethered to the cytosol. Once NF-kB is able to translocate to the nucleus, it binds to promoter regions of several pro-inflammatory genes and leads to an inflammatory response.

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
