# Peer review of "Heat Shock Protein 70 (HSP70) Induction: Chaperonotherapy for Neuroprotection after Brain Injury"

_cells, 2020, doi:10.3390/cells9092020_

Round 1

Reviewer 1 Report

The rewrite was responsive to the previous critiques and the manuscript has been improved. This is a timely and informative review.

Author Response

Dear Reviewer,

Thank you for your positive comments and overall review. With your help, we hope that this paper has become closer to seeing good results.

Sincerely,

Reviewer 2 Report

Although the Authors have raised an interesting topic, the general impression after reading the manuscript is rather unfavourable. The work is written a bit chaotic and requires thorough corrections, including linguistic ones. I'm afraid that in its current form the manuscript is not suitable for publication. Below is the justification.

  • Errors in style, grammar; some examples are listed below;

L16 “Their chaperone properties can assist in protein”  instead of “Their” it should be “Its”

L19-21 “Through the use of  mutant animal models, gene transfer or heat stress, a number of studies have now reported positive outcomes of HSP70.”

This sentence is not complete. What do Authors mean by writing “positive outcomes of HSP70”? Do they mean “positive outcomes of HSP70 induction”?

L35-36 “When core body temperature exceeded 40oC, increased transcripts of HSPs protein were detected in postmortem brain specimens.”  Instead of “transcripts of HSPs protein” it should be “transcripts of HSPs”

L51-52 “As mentioned above, multiple disciplines have studied several of these compounds, which utilized their ability to block HSP90 leading to induction of HSP70 [8,9].” What kind of compounds?

  • Introduction. The nature of "brain injury" should be clearly defined.
  • L31-33 There is a missing reference to this part of the manuscript. The reference that appears later in the text (Doberentz, E. et al. 2017) does not contain relevant information.
  • In Chapter “Classification and functional roles of Heat shock proteins” there is no sufficient information concerning the classification of the HSPs family. What about other HSP subfamilies? The authors in the following chapters devote a lot of space to Hsp27 protein, not to mention the sHSPs subfamily.
  • L78-80 “Of note, HSP90 functions in tandem with many additional chaperones, to include HSP70 and HSP40, and also co-chaperones including co-chaperones containing TPR, to refold many denatured proteins [18].” TPR abbreviation should be explained.
  • L100-102 “HSP transcription is regulated by the transcription factor-heat shock factors (HSF), which translocated into the nucleus where it interacted with conserved heat shock elements (HSEs) to upregulate genes coding for the inducing HSPs.” Instead of “which translocated into the nucleus” should be “which is translocated into the nucleus”; instead of “where it interacted” should be “where it interacts”; instead of “factor-heat shock factors (HSF)” should be “factor-heat shock factors (HSFs)”
  • L111 “(HSE)” Repeated abbreviation.
  • L116-119 “The ATPs, HSP40 and HSP90, along with freshly synthesized HSP70 protein bind denatured proteins and contribute to repairing the cell by acting as molecular chaperone during which it refolds and trafficks damaged proteins throughout the cell, where it undergoes multiple iterations of refolding the proteins.” Due to grammatical and stylistic errors, this sentence is incomprehensible.
  • The reference to Figure 1.B should be placed earlier in the main text, for example, into the line 129.
  • Figure 1. description does not contain a reference to its specific parts (A and B).
  • Figure 1. The presentation of the data in the figure is illogical. Since Hop and CHIP compete for HSP70’s C-terminus, while Hip and Bag-1 compete for the ATPase domain they should be located in the same schematic binding sites marked on HSP70. In other words, mentioned proteins should be shown in the diagrams, in the same way, i.e. HIP and Chip on the left, HOP, and Bag-1 on the right.
  • L163 CA3 abbreviation should be explained
  • Lack of consistency in the use of abbreviations, sometimes all letters are capitalized, sometimes not (e.g. HIP/Hip; Bcl-2/BCL2).
  • Figure 2. Golgi Apparatus is shown in the figure, whereas the endoplasmic reticulum is mentioned in the main text.
  • Figure 2. Apoptosome is composed of Cyt C, Apaf-1, and Caspase-9 it should be presented differently.
  • Figure 2. There is no need to present apoptosis via sickle-shape figures.
  • Figure 2. An arrow indicating the release of Smac/DIABLO from mitochondrion should be shown.
  • L238 PKB abbreviation should be explained.
  • L267 Instead of “messenger RNAs (mRNA)” should be “messenger RNAs (mRNAs),”
  • L302 LPS abbreviation should be explained.
  • L306 NADPH abbreviation should be explained.
  • Figure 3. Cytosol should be marked. Does the brain injury is triggered by extracellular- or intracellular factors?
  • Figure 3. Description. The composition of NF-kB should mention should be mentioned earlier.
  • L329 MMP-9 abbreviation should be explained.
  • Suggestion. The chapter's title "Pharmacological Therapy of Heat Shock Protein 70 in Brain injury" would be more appropriate after modification, e.g. "Heat Shock Protein 70 as a (pharmacological) therapeutic target for brain injury"

Author Response

Dear Reviewer

 We thank the reviewers for fruitful advice, especially for suggesting better terms and sentences. We have revised the manuscript Cells-866452 on the basis of the referees’ comments.

 Hopefully, you will consider that our manuscript is now suitable for publication in the “Cells”.

Sincerely yours,

Round 2

Reviewer 2 Report

The manuscript has been significantly improved. However, it still needs minor corrections. For example, the shape used in Figures 2 and 3 to represent Hsp70 should be identical. In Figure 2, an additional arrow should be placed in the direction from "Brain injury" to "Fas". Moreover, proofreading of text by the native speaker is highly recommended.

Author Response

Dear Reviewer

 We thank the reviewers for fruitful advice, especially for suggesting better terms and sentences. We have revised the manuscript Cells-866452 on the basis of the referees’ comments. We also made the English correction throughout professional editors at Editage.

 Hopefully, you will consider that our manuscript is now suitable for publication in the “Cells”.

Sincerely yours,

This manuscript is a resubmission of an earlier submission. The following is a list of the peer review reports and author responses from that submission.

Round 1

Reviewer 1 Report

In this manuscript, J.Y.Kim et al. summarizes some neuroprotective mechanisms of HSP70 and they also attempt to discuss potential manners for the use of this chaperone as a pharmacological target to ultimately improve neurological outcomes in acute neurological diseases. The latter aim was not achieved at all. The main criticism is that the article is not providing substantial information about the field, and several times, the statements are quite questionable.

Along the manuscript the chaperone is called HSP70, but in the first paragraph it is named according to the new nomenclature for its gene, HSPA1B. This should be clarified for readers who are not familiarized with the field. Also, the gene should be shown in italics.

Lines 39-40:  “HSPs seem to act like cytosolic chaperones” is not a happy sentence, they ARE chaperones and not always they are working as “cytosolic” proteins.

Line 78: “HSP90 binds proteins while in an inactive state. They become activated once dissociated from HSP90”  This is a terrible statement! There are client-proteins that are inactive when Hsp90 is dissociated (for example Tyr-kinases). Steroid receptors do require Hsp90 to bind the hormone otherwise the receptor is not activated, and the Hsp90-based heterocomplex associated to steroid receptors is indeed required for the transport of the receptor towards the nucleus. Actually, this mechanism is not exclusive for steroid receptors but it is also required for other Hsp90-client proteins.

Line 100:  “The chaperone functions of HSP70 and HSP90 appear to have opposite effects”. The most common feature of both chaperones is to work in a coordinated manner rather than to opposite one another! The example given here related to protein degradation is not the best one.

Line 111:  “The unique activity of the HSF1 binding to  the HSP genes’ 5’promoter region is the molecular vehicle for the regulation of HSP70 induction”.  ????  Quite remarkable statement!! HSF1 does not do anything else!  By the way, the worthy description of the hypothesis by which HSF1 is activated is simply ... too elemental and imprecise. There are several theories in this regard that have been simply ignored.

The authors describe apoptotic pathways and some interactions of Hsp70 with key components of the pathways, but there is no analysis of the type and quality of such interactions, the description is extremely superficial. Because this article is supposedly based on the neuroprotective action of Hsp70, it results unclear the specific role of Hsp70 in neurons (if any) versus non nervous cells.

Suddenly, the article is focused on Hsp90 inhibitors without marking a clear nexus with Hsp70. Perhaps this matter was addressed due to the fact that in general, these ligands induce Hsp70 expression. Nonetheless, this implication is not explained or supported, and the potential reasons for discussing this topic are absent.

It is also stated that the use of geldanamycin failed at the clinical level due to its poor solubility in water. Even though its low solubility is indeed a problem, the clinical failure was due to its extremely high hepato-and nephro-toxicity, which are not even mentioned here.

Remarkably, the entire field of small molecules capable to target Hsp70 is absent. Combinations of N-terminal or C-terminal ligands with other HSP ligands (in particular Hsp90 and Hsp27) is a field itself, but it is not even mentioned in this article. Nonetheless, in the abstract it was stated that the article "discusses potential ways in which this class of endogenous therapeutic molecules could be practically translated by  pharmacological means to ultimately improve neurological outcomes in acute neurological disease." 

The potential mechanisms proposed to explain the effects of extracellular Hsp70 are absent.

Hsp70 was first observed to be induced in brain regions that were relatively resistant to ischemic insults. The reasons for this are not addressed in this article, whose primary focus is Hsp70 in brain injury.

Author Response

Dear Reviewer

Acording to the reviewer 1's suggestions and recommenadtion we have throughly changed the information and description of the manuscript. We believe that the present format of the manuscript posess a  good impact in the field neuroprotection in brain injury regarding Hsp70.

Sincerely yours,

Reviewer 2 Report

This manuscript reviews the latest findings on HSP70's modes of action, particularly in conjunction with HSP90. It reviews the effects of HSP70 on several systems ranging from intracellular to immunological. It further summarizes studies on drugs that affect the HSP70 system. Overall, this is a well-written review that summarizes findings in an informative fashion without getting bogged down in extraneous details. It lays out clearly the present state of the field with regard to HSP70 and possible pharmacological manipulations. The review is expected to be of use to workers in the relevant field. In sum, this is an excellent and timely review that is expected to help move the field forward.

MAJOR COMMENTS

  1. The discussion of the integrated action of HSP70 and HSP90 was excellent. The discussions about pharmacological manipulations was also excellent and hopefully will help accelerate work in this area.

MINOR COMMENTS

  1. The use of the word "seems" in the first paragraph is inappropriate. The word "seems" implies the possibility that the fact is different from stated. However, the stated facts are well-established. Please change the word "seems" to a more appropriate word.
  2. I recommend going through and giving the manuscript one more good edit. I found a few typos here and there.

Author Response

(The authors gave the same response as above.)

Round 2

Reviewer 1 Report

In this revised version, the authors have deleted or modified most of the statements that were conceptually wrong or incorrectly addressed. Nonetheless, the main criticism still stands: in the best-case scenario, it adds too little to the field.

The basic concepts addressed are known since several years ago. Novel, contrasting or challenging topics are not included. Importantly, the logic to connect the actions of Hsp70 and Hsp90 is poorly accomplished, the use of specific drugs against Hsp70 and their consequences is a missing topic, there is no link between Hsp90 inhibitors and the main focus of the article, etc.  

In summary, the manuscript looks like a scholarly monograph rather than a review article where precise updates on the most recent advances and developments in the field are discussed and profoundly analyzed.